# Subgrid-scale variability of clear-sky relative humidity and forcing by aerosol-radiation interactions in an atmosphere model

Paul Petersik[1], Marc Salzmann[1], Jan Kretzschmar[1], Ribu Cherian[1], Daniel Mewes[1], and Johannes Quaas[1]

[1]Leipzig Institute for Meteorology, Universität Leipzig

*Correspondence to:* Paul Petersik (paul.j.petersik@gmail.com)

**Abstract.** Atmosphere models with resolutions of several tens of kilometres take subgrid-scale variability of the total specific humidity $q_t$ into account by using a uniform probability density function (PDF) to predict fractional cloud cover. However, usually only mean relative humidity, $\overline{\mathrm{RH}}$, or mean clear-sky relative humidity, $\overline{\mathrm{RH}}_\mathrm{cls}$, is used to compute hygroscopic growth of soluble aerosol particles. While previous studies based on limited-area models and also a global model suggest that sub-grid scale variability of RH should be taken into account for estimating RFari, here we present the first estimate of RFari using a global atmospheric model with a parameterization for sub-grid scale variability of RH that is consistent with the assumptions in the model. For this, we sample the subsaturated part of the uniform RH-PDF from the cloud cover scheme for its application in the hygroscopic growth parameterization in the ECHAM6-HAM2 atmosphere model. Due to the non-linear dependence of the hygroscopic growth on RH, this causes an increase in aerosol hygroscopic growth. Aerosol optical depth (AOD) increases by a global mean of $0.009$ ($\sim 7.8\,\%$ in comparison to the control run). Especially over the tropics AOD is enhanced with a mean of about $0.013$. Due to the increase in AOD, net top of the atmosphere clear-sky solar radiation, $SW_{\mathrm{net,cls}}$, decreases by $-0.22\,\mathrm{W\,m^{-2}}$ ($\sim -0.08\,\%$). Finally, the radiative forcing due to aerosol-radiation interactions (RFari) changes from $-0.15\,\mathrm{W\,m^{-2}}$ to $-0.19\,\mathrm{W\,m^{-2}}$ by about $31\,\%$. The reason for this very disproportionate effect is that anthropogenic aerosols are disproportionally hygroscopic.

## 1 Introduction

Aerosols have a significant impact on the climate system by interacting with radiation and clouds. Solar and thermal radiation interact with aerosols by absorption and scattering processes. Despite extensive research on atmospheric aerosols, the effective radiative forcing due to aerosol-radiation interactions (ERFari) has still a large uncertainty. The ERFari combines effects from radiative forcing due to aerosol-radiation interactions (RFari) and rapid adjustments and is estimated to be $-0.45$ ($-0.95$ to $+0.05$) $\mathrm{W\,m^{-2}}$ by the 5th assessment report (AR5) of the Intergovernmental Panel on Climate Change (IPCC) (Boucher et al., 2013). The radiative forcing by aerosol-radiation interactions from sulphate ($-0.4\,\mathrm{W\,m^{-2}}$) and nitrate ($-0.11\,\mathrm{W\,m^{-2}}$) is a cooling effect on the radiative balance of the Earth due to increased scattering of solar radiation. In contrast, black carbon ($+0.4\,\mathrm{W\,m^{-2}}$) is warming the Earth's climate due to absorption of solar radiation. Additionally, it is uncertain if primary and secondary organic aerosols, aerosols from biomass burning and mineral dust have a net cooling or a warming effect (e.g. Bond

et al., 2013; Shindell et al., 2013; Myhre et al., 2013; Ginoux, 2017). In total, anthropogenic aerosols have very likely a cooling effect through aerosol-radiation interactions on the radiative balance of the Earth (Boucher et al., 2013).

Through non-linear relationships described by Beer-Lambert's law (Lambert, 1760; Beer, 1852) and Mie scattering (Mie, 1908), the extinction of radiation is related to the aerosol particle radius. The aerosol particle radius of hygroscopic aerosols like sulphate or sea salt aerosols increases in a humid environment due to attraction of water. This hygroscopic growth is a non-linear function of the ambient relative humidity (RH), where hygroscopic growth is especially enhanced close to saturation (Köhler, 1936). Therefore, extinction due to hygroscopic aerosols increases strongly when the humidity of the ambient air approaches saturation (Zieger et al., 2013; Skupin et al., 2016; Haarig et al., 2017).

It is known that humidity varies on subgrid scales in general circulation models (GCMs) with largest subgrid-scale variability in the middle troposphere (e.g. Quaas (2012)). However, GCMs that just use the grid-box mean relative humidity $\overline{RH}$ to calculate hygroscopic growth of aerosols do not take this subgrid-scale variability of humidity and its effect on radiation into account. Studies based on limited-area models suggest that GCMs may underestimate the RFari of sulphate aerosols by 30 to 80% when not considering subgrid-scale variability of RH for the hygroscopic growth of sulphate particles (Haywood et al., 1997; Petch, 2001; Myhre et al., 2002). These studies use high resolution models and compare the results from calculations of radiative forcing that keep the high resolution of RH with either calculation where RH is averaged spatially beforehand to mimic a GCM resolution or results from model configurations with a coarser resolution. In addition, recent studies show that models with a coarse resolution which do not take subgrid-scale variability of various aerosol properties into account underestimate aerosol radiative forcing (Gustafson et al., 2011) and have a significant negative bias in aerosol optical depth (Weigum et al., 2016).

First attempts to implement a subgrid-scale variability of RH in a GCM for the calculation of RFari by sulphate were made by Haywood and Shine (1997) and Haywood and Ramaswamy (1998). However, these studies make strong simplifications about the shape of the used probability density function (PDF) and are not consistent with the cloud cover scheme. Haywood and Shine (1997) prescribe the RH distribution globally for clear skies. Here, for each grid cell and height level five fixed RH-values are taken from a normal distribution around RH= 70 %. They find that RFari by sulphate is 24% greater than the non-hydrated forcing when using the grid-box mean RH. However, RFari by sulphate increases up to 37% when the subgrid-scale variability of RH is applied and the correlation between clouds and areas of high relative humidity is taken into account. Hence, RFari by sulphate increased by about 10% from simulations that use grid-box mean RH to simulations with a subgrid-scale variability of RH. Haywood and Ramaswamy (1998) have a more sophisticated approach. They use a triangular shaped relative humidity distribution around the grid-box mean RH with a magnitude of $\pm 10\%$ that is truncated at RH= 1.0 as proposed by Haywood et al. (1997). They show that RFari by sulphate is enhanced by 9% due to the subgrid-scale variability of RH when clouds are included. We want to point out that Haywood and Ramaswamy (1998) do not consider variations of width and shape of the used distribution. This is a rather strong simplification (especially having the non-linear hygroscopic growth in mind) in comparison to findings of Quaas (2012) who suggests a change of the width of a uniform PDF from about $\pm 20\%$ at the surface to about $\pm 65\%$ in the middle troposphere.

In this study, we implement a stochastic parameterization of subgrid-scale variability of clear-sky relative humidity $RH_{cls}$ into the global aerosol-climate model of the Max Planck Institute for Meteorology (MPI-M) ECHAM6-HAM2 (Zhang et al., 2012; Stevens et al., 2013). For this, we use a uniform probability density function (PDF) that reproduces the subsaturated part of the cloud cover scheme from Sundqvist et al. (1989) that is used by ECHAM6 (Stevens et al., 2013). The width of the PDF

from the cloud cover scheme is a function of height (Quaas, 2012; Rosch et al., 2015). Hence, our parameterization inherits this feature. ECHAM6-HAM2 until now used the grid box-mean clear-sky relative humidity $\overline{RH}_{cls}$ to calculate hygroscopic growth (Zhang et al., 2012). Now, rather than using the grid-box mean, the PDF of the subgrid-scale variability of $RH_{cls}$ is randomly sampled for each time step, grid cell and height level to compute the growth factor $gf$ (see section 2). Hence, the parameterization complies with the necessity to be consistent and to introduce as few as possible empirical/tunable parameters.

For a more elaborate discussion on the topic in the literature we refer to e.g. Arakawa (2004). Furthermore, hygroscopic growth is computed in ECHAM6-HAM2 for all hygroscopic aerosol constituents that are incorporated in the model. Therefore, the effect of subgrid-scale variability of $RH_{cls}$ on hygroscopic growth is included for all hygroscopic aerosol particles in the model.

A very similar method of subgrid-scale variability of humidity is for example applied on the convective scheme of the European Centre for Medium-Range Weather Forecasts (ECMWF) ensemble prediction system by Tompkins and Berner (2008).

They show that their new stochastic convective scheme generally improves the skill of the operational system for most variables in the short to medium range in the mid-latitudes. More generally, we want to emphasize that stochastic parameterizations are not only a method to estimate uncertainties but lead to a better representation of the mean state of the atmosphere. This was recently summarized in Berner et al. (2016).

In section 2 of this article we describe the aerosol module HAM2 in more detail and introduce our stochastic parameterization

of clear-sky relative humidity. Then, we investigate in section 3 how the new parameterization changes optical and radiative properties of the atmosphere. Afterwards, the results are discussed in section 4. Finally, this study is summarized with an outlook for further research in section 5.

## 2   Methods

### 2.1   Aerosol Module HAM2

In this section, we briefly describe the aspects of the micro-physical aerosol module HAM2 (Zhang et al., 2012) that are relevant for this study. The micro-physical aerosol module HAM2 is the successor of its first version that was introduced by Stier et al. (2005). HAM2 is built as an extension of the atmospheric general circulation model ECHAM6 (Stevens et al., 2013). It incorporates the following aerosol components: Sulphate (SO4), black carbon (BC), organic carbon (OC), sea salt (SS) and mineral dust (DU). The module predicts the evolution of the aerosol population based on 7 log-normal modes (4 soluble (S)

and 3 insoluble (I)) that describe the size distribution of atmospheric aerosol. The modes are divided into Nucleation (N, $r < 0.005\,\mu m$), Aitken (K, $0.005 < r < 0.05\,\mu m$), Accumulation (A, $0.05 < r < 0.5\,\mu m$) and Coarse (C, $r > 0.5\,\mu m$) mode and are abbreviated in the following such as CS for soluble Coarse mode. The main soluble aerosol constituents are SS, SO4 and

OC. DU and BC are considered as insoluble on emission. However, insoluble aerosols can become soluble if they merge with soluble particles due to internal mixing by ageing processes such as condensation and coagulation (Vignati et al., 2004).

Aerosol radiative properties are calculated for 24 spectral bands for shortwave and 16 bands for longwave radiation using Mie theory by applying the algorithm suggested by Toon and Ackerman (1981). The model uses the volume-weighted average refractive indices for internally mixed aerosols where aerosol water is included (Stier et al., 2005; Ghan and Zaveri, 2007). The effective complex radiative indices and the Mie size parameter are then used for the aerosol radiative properties, namely extinction cross section, single scattering albedo, and asymmetry parameter in the radiation scheme. For the version 2 of the aerosol module the refractive indices for black carbon were updated with values from Bond and Bergstrom (2006) that led to a reduction of the negative bias due to aerosol absorption enhancement (Stier et al., 2007). In contrast to Jacobson (2012) and Bond et al. (2013), HAM2 does not include a very strong absorption enhancement for absorbing particles inside clouds. It should be noted that the hypothesis of Jacobson (2012) is very controversial and not supported by most other studies (e.g. Twohy et al., 1989; Chýlek et al., 1996; Liu et al., 2002).

## 2.2 Hygroscopic growth in HAM2

Soluble particles can grow in size due to the attraction of water. This hygroscopic growth can be described by the growth factor $gf = r_{\mathrm{wet}}/r_{\mathrm{dry}}$, where $r_{\mathrm{wet}}$ and $r_{\mathrm{dry}}$ are the wet and dry radius of an aerosol particle, respectively. Petters and Kreidenweis (2007) introduced the $\kappa$-Köhler theory to calculate the growth factor as a function of relative humidity and temperature

$$\frac{\mathrm{RH}}{\exp\left(\frac{A_{\mathrm{K}}(T)}{D_{\mathrm{dry}}gf}\right)} = \frac{gf^3 - 1}{gf^3 - (1 - \kappa)} \tag{1}$$

where RH is the ambient relative humidity, $D_{\mathrm{dry}}$ the dry diameter, $A_{\mathrm{K}}$ the temperature-dependent parameter of the Kelvin (curvature) effect and $\kappa$ the hygroscopicity. A $\kappa$ value of 0 describes completely hydrophobic aerosols, whereas $\kappa$ values greater than 0.5 describe very hygroscopic aerosols. Aerosol constituents with very high $\kappa$ values in HAM2 are sulphate and sea salt (see Table 1). Note that the $\kappa$ value for sulphate in HAM2 is in the range of the observed value for ammonium sulphate (0.33 - 0.72) (Petters and Kreidenweis, 2007). Furthermore, HAM2 does not include nitrate in its current set-up. The hygroscopicity of internally-mixed aerosols is determined by calculating the volume-weighted average of the $\kappa$-values form each soluble compound. In Eq. (1) $gf$ is a strictly-monotonically increasing, non-linear function with positive curvature for $\mathrm{RH} \in [0,1]$.

ECHAM6-HAM2 uses the grid-box mean clear-sky relative humidity $\overline{\mathrm{RH}}_{\mathrm{cls}}$ in Eq. (1) to calculate the hygroscopic growth for each aerosol mode. $\overline{\mathrm{RH}}_{\mathrm{cls}}$ is chosen, instead of grid-box mean relative humidity $\overline{\mathrm{RH}}$, because $\overline{\mathrm{RH}}_{\mathrm{cls}}$ is a better estimate for the relative humidity in the cloud-free part of a grid cell than $\overline{\mathrm{RH}}$ and cloud processing and cloud radiative effects are dominant in the cloudy part of a grid box as reasoned in Stier et al. (2005) for ECHAM5-HAM1. This means that since the aerosol-radiation interactions have a small effect compared to the cloud reflectivity in the cloudy part of a grid cell, swelling is only approximately treated in the cloudy part. $\overline{\mathrm{RH}}_{\mathrm{cls}}$ is diagnosed from predicted $\overline{\mathrm{RH}}$. For this, saturation is assumed in clouds (RH $= 1$). When the grid-box is cloud-free ($f = 0$) or partly cloudy ($0 < f < 1$), clear-sky relative humidity is computed by $\overline{\mathrm{RH}}_{\mathrm{cls}} = \left(\overline{\mathrm{RH}} - f\right)/(1 - f)$ where $f$ is the fractional cloud cover. For overcast grid-boxes ($f = 1$), the clear-sky relative

**Table 1.** $\kappa$-values as used in HAM2 from Zhang et al. (2012). The $\kappa$-value for sulphate in HAM2 is in the range of the observed value for Ammonium sulphate (Petters and Kreidenweis, 2007). Furthermore, HAM2 does not include nitrate in its current set-up.

| Compound | $\kappa$ |
|---|---|
| Sulphate | 0.60 |
| Sea salt | 1.12 |
| Primary organic aerosol | 0.06 |
| Secondary organic aerosol | 0.022 - 0.070 |
| Black carbon | 0 |
| Mineral dust | 0 |
| Nitrate | - |

humidity is set to saturation as well ($\overline{\text{RH}}_{\text{cls}} = 1$). Using the $\overline{\text{RH}}_{\text{cls}}$ in Eq. (1) implies that no subgrid-scale variability of $\text{RH}_{\text{cls}}$ is used beyond the information supplied by the fractional cloud cover.

## 2.3 Stochastic parameterization for the subgrid-scale variability of clear-sky relative humidity

In the ECHAM model, subgrid-scale variability of specific humidity is already used for the prediction of fractional cloud cover
(Stevens et al., 2013). The cloud scheme assumes a uniform probability density function (PDF) as proposed by Sundqvist et al. (1989) for the horizontal subgrid-scale variability of total-water $q_{\text{t}}$ between $\bar{q}_{\text{t}} - \Delta q$ and $\bar{q}_{\text{t}} + \Delta q$, where $\Delta q = \gamma q_{\text{s}}$ (see Fig. 1a). $\bar{q}_{\text{t}}$ is the model-predicted grid-box mean total-water specific humidity and $q_{\text{s}}$ the saturation specific humidity computed from the predicted grid-box mean temperature. The scaling parameter $\gamma$ is varying in the vertical but otherwise assumed to be constant in space and time. It can be calculated by $\gamma = 1 - \text{RH}_{\text{crit}}$, where $\text{RH}_{\text{crit}}$ is the critical relative humidity that is a
function of height (Quaas, 2012; Rosch et al., 2015). Fractional cloud cover occurs if the grid box mean relative humidity $\overline{\text{RH}}$ exceeds the critical relative humidity. In ECHAM6-HAM2, $\text{RH}_{\text{crit}}$ is parametrized as a by

$$\text{RH}_{\text{crit}}(p) = c_{\text{t}} + (c_{\text{s}} - c_{\text{t}}) \exp\left[1 - \left(\frac{p_{\text{s}}}{p}\right)^{n_{\text{x}}}\right] \tag{2}$$

with $p$ the ambient pressure and $p_{\text{s}}$ the surface pressure. Furthermore, $c_{\text{t}} = 0.7$ and $c_{\text{s}} = 0.9$ are the critical relative humidity values at the top of the atmosphere (TOA) and the surface, respectively, and $n_x = 4$. The given values for $c_{\text{t}}$, $c_{\text{s}}$ and $n_x$ are
the same as in the in the cloud cover scheme of ECHAM6 in the standard set-up. Note, that satellite observations analysed by Quaas (2012) suggest a considerably stronger vertical change with $c_{\text{t}} = 0.34$ in stable regions and $c_{\text{t}} = 0.37$. However, we keep the values for $c_{\text{t}}$, $c_{\text{s}}$ and $n_x$ as in the standard set-up to ensure the comparability with older studies. While the formulation of $\text{RH}_{\text{crit}}$ is specific to the ECHAM6 model, the cloud cover scheme from Sundqvist et al. (1989) has also been applied in other global models. Furthermore, the Tiedtke (1993) cloud cover scheme which is for example used in the Geophysical Fluid
Dynamics Laboratory (GFDL) atmosphere model AM3 (Donner et al., 2011) assumes a uniform PDF of total water as well.

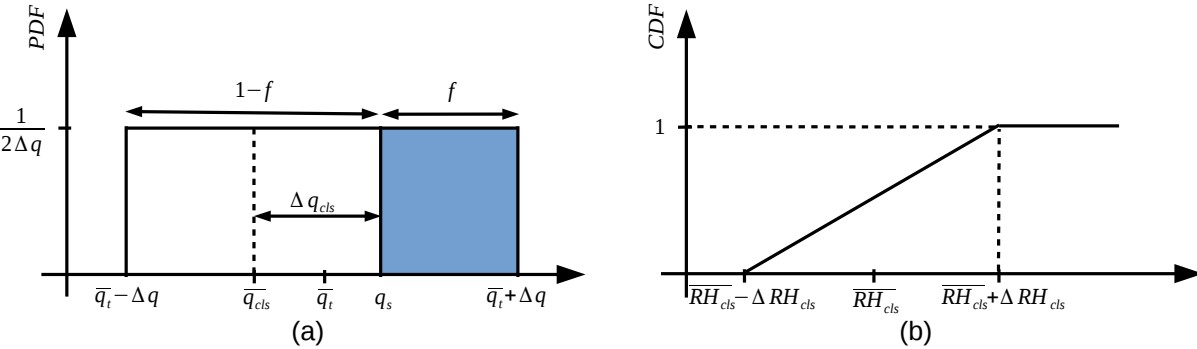

**Figure 1.** (a) Scheme for predicting fractional cloud cover, $f$, with a uniform PDF for the total-water $q_t$. The blue area indicates the fraction of a grid cell which is covered by clouds. $\Delta\mathrm{RH}_{\mathrm{cls}}$ is set to $\Delta q/q_s = 1 - \mathrm{RH}_{\mathrm{crit}}$ for a PDF around $\overline{\mathrm{RH}}_{\mathrm{cls}}$ if no fractional cloud cover is present (not depicted) or set to $(q_s - q_{\mathrm{cls}})/q_s = 1 - \overline{\mathrm{RH}}_{\mathrm{cls}}$ if fractional cloud cover is present (in the figure shown in terms of specific humidity). (b) Cumulative distribution function (CDF) for a uniform PDF around $\overline{\mathrm{RH}}_{\mathrm{cls}}$. By the inversion of the CDF and with a random number $a \in [0,1]$ (see Eq. (6)), a value between $\mathrm{RH}_{\mathrm{cls}} - \Delta\mathrm{RH}_{\mathrm{cls}}$ and $\mathrm{RH}_{\mathrm{cls}} + \Delta\mathrm{RH}_{\mathrm{cls}}$ is sampled and used as the argument for the hygroscopic growth.

Several global atmosphere models including ECHAM6-HAM2 already make assumptions to account for the subgrid-scale variability of other atmospheric variables, e.g. for vertical velocity when computing droplet activation rates (Ghan et al., 1997; Lohmann et al., 2007; Golaz et al., 2011). However, subgrid-scale variability of RH or $\mathrm{RH}_{\mathrm{cls}}$ is not taken into account when computing hygroscopic growth of interstitial aerosols besides in some studies that made strong simplification regarding the shape and variation of the used PDF as explained in the introduction (Haywood et al., 1997; Haywood and Ramaswamy, 1998).

For our stochastic parameterization of subgrid-scale variability of $\mathrm{RH}_{\mathrm{cls}}$, we use the subsaturated part of the $q_t$-PDF from the cloud cover scheme in not-overcast cases (see Fig. 1a). This diagnosed PDF is transformed into a $\mathrm{RH}_{\mathrm{cls}}$-PDF dividing it by $q_s$. Afterwards, it is sampled for the stochastic parameterization of subgrid-scale variability of $\mathrm{RH}_{\mathrm{cls}}$. The width of the $q_t$-PDF in the cloud cover scheme is:

$$2\Delta q = 2\gamma q_s = 2 \cdot (1 - \mathrm{RH}_{\mathrm{crit}})q_s \tag{3}$$

Dividing Eq. (3) by $q_s$ yields the width of the corresponding RH-PDF. For cloud-free grid-boxes this RH-PDF is equivalent to the $\mathrm{RH}_{\mathrm{cls}}$-PDF. In this case, its which width is

$$2\Delta\mathrm{RH}_{\mathrm{cls}} = 2\frac{\Delta q}{q_s} = 2 \cdot (1 - \mathrm{RH}_{\mathrm{crit}}). \tag{4}$$

$\mathrm{RH}_{\mathrm{crit}}$ is computed by Eq. (2). However, when fractional cloud cover is present $\Delta\mathrm{RH}_{\mathrm{cls}}$ has to be adjusted to

$$\Delta\mathrm{RH}_{\mathrm{cls}} = \frac{q_s - q_{\mathrm{cls}}}{q_s} = 1 - \overline{\mathrm{RH}}_{\mathrm{cls}} \tag{5}$$

so that the variation of $\mathrm{RH}_{\mathrm{cls}}$ occurs in the subsaturated part of the cloud cover PDF (see Fig. 1a).

Afterwards, instead of using $\overline{\mathrm{RH}}_{\mathrm{cls}}$ as input for the calculation of the $gf$, a stochastic value for clear-sky relative humidity, $\mathrm{RH}_{\mathrm{cls,new}}$, from the inversion of the cumulative distribution function (CDF) is drawn (see Fig. 1b). For this, a random number, $a \in [0,1]$, is generated and inserted into the following equation:

$$\mathrm{RH}_{\mathrm{cls,new}} = \overline{\mathrm{RH}}_{\mathrm{cls}} + \Delta\mathrm{RH}_{\mathrm{cls}}(2a - 1). \tag{6}$$

Note, that the integration of the model is not fully deterministic in the current setting. If one preferred a deterministic model, one could configure the random number generator such that in reach integration the same random numbers are generated

## 2.4   Model settings and postprocessing

ECHAM-HAM2 is run with a resolution of T63L31. For aerosol emissions, the AEROCOM II data for 1850 for pre-industrial (PI) and for 2000 for present-day (PD) simulations are used (Lamarque et al. (2010); see section 7). Climatological sea surface
temperature (SST) and sea ice distributions are prescribed. Ten-year model free-running (no nudging) simulations starting 1 January 2000 are performed with PI and PD aerosol emissions, both with and without the new parameterization. The total ERF by anthropogenic aerosols, ERFaer, is computed by

$$\mathrm{ERFaer} = (SW_{\mathrm{net}} + LW_{\mathrm{net}})_{PD} - (SW_{\mathrm{net}} + LW_{\mathrm{net}})_{PI} \tag{7}$$

where the short and longwave radiative fluxes, $SW$ and $LW$, are at the top of atmosphere (TOA). The radiative forcing due to
aerosol-radiation interactions, RFari, is computed as suggested by Ghan (2013):

$$\mathrm{RFari} = (SW_{\mathrm{net}} - SW_{\mathrm{net,clean}})_{PD} - (SW_{\mathrm{net}} - SW_{\mathrm{net,clean}})_{PI} \tag{8}$$

Again, radiative fluxes are at TOA. The subscript clean indicates the results of the radiative transfer equation for an atmosphere with no aerosols.

To depict changes in hygroscopic growth we define the squared ratio

$$\beta = \left( \frac{gf_{\mathrm{stoch}}}{gf_{\mathrm{control}}} \right)^2 \tag{9}$$

where $gf_{\mathrm{stoch}}$ and $gf_{\mathrm{control}}$ account for the growth factor in the model run with the stochastic parameterization and the control model run, respectively. The squared ratio scales with the effective extinction cross section and therefore describes the influence on aerosol optical depth (AOD).

Satellite retrievals of AOD from the MODerate Resolution Imaging Spectroradiometer (MODIS) platform Aqua (Levy et al.,
2013) from the period between August 2002 and December 2010 are used to evaluate the results of implementing subgrid-scale variability of $\mathrm{RH}_{\mathrm{cls}}$ into the model. The temporal mean values of AOD measurements ($\overline{\mathrm{AOD}}_{\mathrm{MODIS}}$) for the entire time span (08/2002 - 12/2010) are compared to the temporal means (01/2000 - 12/2009) of the model data ($\overline{\mathrm{AOD}}_{\mathrm{control}}$, $\overline{\mathrm{AOD}}_{\mathrm{stoch}}$).

## 3 Results

In the following results from PD simulations, if not specified differently, are presented. We compute uncertainties for a 95 %-confidence interval on basis of yearly mean values from the temporal variability. In differences, uncertainties are added in quadrature.

In Fig. 2a the global mean profiles of $\beta$ are shown for all soluble aerosol mode. Hygroscopic growth of aerosols is in general enhanced due to the implementation of a subgrid-scale variability of $RH_{cls}$. We find that the effect is stronger for aerosol particles with a large particle radius. Thus, the effect is strongest for particles from the CS mode (red line in Fig. 2a) and weakest for particles from the NS mode (black line in Fig. 2a). Moreover, the effect on hygroscopic growth has a maximum between 700 hPa and 600 hPa for each aerosol mode.

The global mean AOD increases by $0.009 \pm 0.002$ ($\sim 7.8\,\%$). The response in AOD is weaker in simulations with PI emissions with a global mean difference of $0.006 \pm 0.002$ ($\sim 6.0\,\%$). Figure 3a shows that the AOD increased especially in lower latitudes with a mean of about 0.013 in the tropics. Furthermore, the figure reveals that the AOD of diagnosed aerosol water (WAT) dominates the change in total AOD, not the change in dry matter of SS or SO4. Figure 3b shows the zonal mean AOD values from the model runs with and without the stochastic parameterization and from satellite measurements of MODIS-Aqua. Changes due to the new parameterization are small in comparison to the general difference between modelled and measured AOD. The absorption aerosol optical depth (AAOD) increased by $0.12 \pm 0.04 \cdot 10^{-3}$ ($\sim 4.7\,\%$), mainly due to an increase in AAOD by BC of about $0.11 \cdot 10^{-3}$. However, note that in absolute terms the change of AOD is nearly two orders of magnitude greater than the change of AAOD.

Furthermore, the implementation of the new parameterization enhanced the ratio of scattering efficiency to total extinction efficiency, $\omega$, for the CS, AS and KS aerosol mode with a maximum for the KS mode ($2.6\,\%$). The effective extinction cross section, $\sigma$, increases for the CS, AS and KS aerosol mode as well. The strongest change is visible for the KS mode with a change by 15.1 %. Note that no output for $\omega$ and $\sigma$ is generated by the model for NS mode. Finally, the Ångström exponent for wet particles, $\alpha$, changes by -0.8 $\cdot 10^{-3}$ $\pm 11.5 \cdot 10^{-3}$ ($-0.11\,\%$). The total cloud cover decreased by $-0.08 \pm 0.14\%$ in PD simulations. In contrast, it increased by $0.17 \pm 0.14$ in PI simulations. The global mean profile of cloud cover $f$ in Fig. 4 reveals a slight increase of cloud cover between 700 and 900 hPa for PD simulations, whereas it mainly decreased below and above this layer. However, in PI runs $f$ increased for most parts of the atmosphere with a very little decrease at around 600 hPa.

In the following, solar and thermal clear-sky radiation represent the idealised solar and thermal irradiance that would arise from an atmosphere where clouds are absent, whereas all-sky stands for the irradiance that takes the effect of clouds into account. The net clear-sky solar radiation $SW_{net,cls}$ decreases by $-0.22 \pm 0.07\,\mathrm{W\,m^{-2}}$. In addition, the net all-sky solar radiation $SW_{net}$ changes by $-0.34 \pm 0.22\,\mathrm{W\,m^{-2}}$. For PI emissions, the effect on $SW_{net,cls}$ is with a change of $-0.13 \pm 0.06\,\mathrm{W\,m^{-2}}$, as expected, smaller than for PD emissions. In contrast, a stronger effect in PI runs than in PD runs is visible for $SW_{net}$ ($-0.47 \pm 0.19\,\mathrm{W\,m^{-2}}$). Responses in the thermal radiation (positive downward) are small. The clear-sky thermal radiation $LW_{net,cls}$ has a slight positive tendency with a mean value of $0.04 \pm 0.09\,\mathrm{W\,m^{-2}}$. Similar to solar radiation, the all-sky thermal radiation $LW_{net}$ changes more than the clear-sky radiation with a global mean of $0.06 \pm 0.14\,\mathrm{W\,m^{-2}}$.

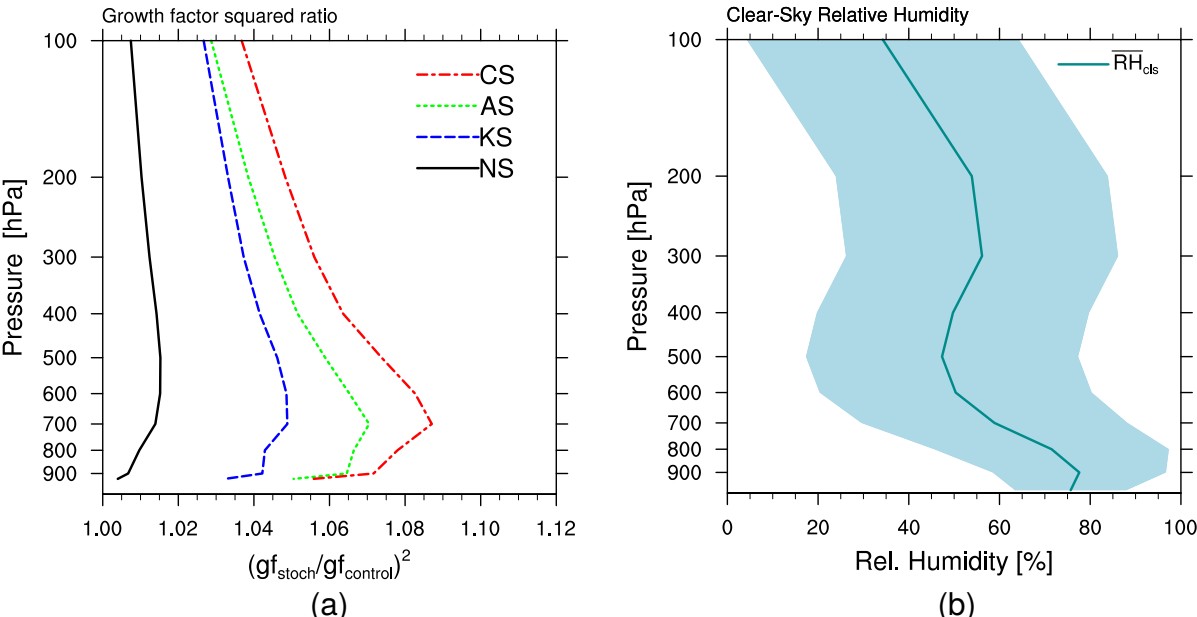

**Figure 2.** (a) Profile of the global mean of the squared ratio of the growth factor between the run with the stochastic parameterization of hygroscopic growth, $gf_{\mathrm{stoch}}$, and the control run, $gf_{\mathrm{control}}$. CS is the soluble Coarse aerosol mode (red). AS the soluble Accumulation (green), KS the soluble Aitken (blue) and NS the soluble Nucleation aerosol mode (black). (b) Profile of global mean clear-sky relative humidity (dark blue line) with its corresponding range of subgrid-scale variability (light blue area).

The comparison of ten-year model runs with PD and PI aerosol emissions reveals a change of the RFari from $-0.15 \pm 0.04\,\mathrm{W\,m^{-2}}$ to $-0.19 \pm 0.04\,\mathrm{W\,m^{-2}}$ (31 %) in runs without and runs with the new parameterization, respectively. This implies that subgrid-scale variability of $RH_{\mathrm{cls}}$ enhances the cooling effect of anthropogenic aerosol emissions by aerosol-radiation interactions in climate simulations. It is interesting to note that the RFari increases substantially given the relatively small impact

5  of the revision on present-day TOA balance. This can be attributed to the fact that anthropogenic aerosol is disproportionally hygroscopic. Furthermore, the effect on RFari translates into the ERF of anthropogenic aerosols (ERFaer) that has as well a negative tendency ($-0.07 \pm 0.27$). A summary of the influence of subgrid-scale variability of $RH_{\mathrm{cls}}$ on optical and radiative variables is given in Table 2.

## 4  Discussion

10  As for the previous section, we discuss in the following the results from PD simulations, if not specified differently. In model runs with the new parameterization aerosol particles swell stronger at each height level due to the non-linear nature of hygroscopic growth (see Eq. 1). The positive curvature of this function for $RH_{\mathrm{cls}} \in [0, 1]$ implies that by applying a uniform PDF

**Table 2.** Changes of global mean values of optical and radiative variables due to the implementation of subgrid-scale variability of $RH_{cls}$ are listed. Uncertainties of the mean value are calculated for a 95 %-confidence interval on basis of yearly mean values from the temporal variability. Uncertainties of differences are added in quadrature. $\omega$ is the ratio of the scattering efficiency to the total extinction efficiency and $\sigma$ the effective extinction cross section. The indices KS, AS and CS indicate Aitken, Accumulation and Coarse mode. $\alpha$ is the wet Ångström exponent, $SW_{net}$ the net short wave radiation and with index $SW_{net,cls}$ the net short wave radiation in the clear-sky part. With the same meaning for the indices $LW$ is the longwave radiation. $TCC$ is the total cloud cover. Results are presented for present-day and pre-industrial emissions.

| Variable | Stoch | Control | Difference | Relative deviation |
|---|---|---|---|---|
| RFari [ W m$^{-2}$] | $-0.19 \pm 0.04$ | $-0.15 \pm 0.04$ | $-0.04 \pm 0.06$ | 31% |
| ERFaer [ W m$^{-2}$] | $-1.52 \pm 0.16$ | $-1.45 \pm 0.21$ | $-0.07 \pm 0.27$ | 5% |

| | Present-Day | | Pre-Industrial | |
|---|---|---|---|---|
| Variable | Difference | Relative deviation | Difference | Relative deviation |
| $SW_{net}$ [ W m$^{-2}$] | $-0.34 \pm 0.22$ | $-0.15\,\%$ | $-0.47 \pm 0.19$ | $-0.20\,\%$ |
| $SW_{net,cls}$ [ W m$^{-2}$] | $-0.22 \pm 0.07$ | $-0.08\,\%$ | $-0.13 \pm 0.06$ | $-0.05\,\%$ |
| $LW_{net}$ [ W m$^{-2}$] | $0.06 \pm 0.14$ | $0.03\,\%$ | $0.26 \pm 0.17$ | $0.10\,\%$ |
| $LW_{net,cls}$ [ W m$^{-2}$] | $0.04 \pm 0.09$ | $0.02\,\%$ | $0.11 \pm 0.10$ | $0.04\,\%$ |
| $TCC$ [ %] | $-0.08 \pm 0.14$ | $-0.13\,\%$ | $0.17 \pm 0.14$ | $0.26\,\%$ |
| $AOD$ | $0.009 \pm 0.002$ | $7.8\,\%$ | $0.006 \pm 0.002$ | $6.0\,\%$ |
| $AOD_{WAT}$ | $0.008 \pm 0.001$ | $9.5\,\%$ | $0.005 \pm 0.001$ | $7.8\,\%$ |
| $AAOD$ | $(0.12 \pm 0.04) \cdot 10^{-3}$ | $4.7\,\%$ | $(0.04 \pm 0.03) \cdot 10^{-3}$ | $2.7\,\%$ |
| $AAOD_{BC}$ | $(0.11 \pm 0.03) \cdot 10^{-3}$ | $5.1\,\%$ | $(0.03 \pm 0.01) \cdot 10^{-3}$ | $3.4\,\%$ |
| $\omega_{KS}$ | $0.015 \pm 0.003$ | $2.64\,\%$ | $0.014 \pm 0.005$ | $2.42\,\%$ |
| $\omega_{AS}$ | $(1.1 \pm 0.4) \cdot 10^{-3}$ | $0.11\,\%$ | $(0.9 \pm 0.6) \cdot 10^{-3}$ | $0.10\,\%$ |
| $\omega_{CS}$ | $(0.03 \pm 0.05) \cdot 10^{-3}$ | $0.003\,\%$ | $(0.05 \pm 0.15) \cdot 10^{-3}$ | $0.005\,\%$ |
| $\sigma_{KS}$ | $(9.7 \pm 1.0) \cdot 10^{-18}$ | $15.1\,\%$ | $(8.0 \pm 1.1) \cdot 10^{-18}$ | $14.3\,\%$ |
| $\sigma_{AS}$ | $(20.8 \pm 6.9) \cdot 10^{-15}$ | $4.5\,\%$ | $(22.7 \pm 7.2) \cdot 10^{-15}$ | $4.8\,\%$ |
| $\sigma_{CS}$ | $(0.69 \pm 0.07 \cdot 10^{-12}$ | $3.9\,\%$ | $(0.71 \pm 0.07) \cdot 10^{-12}$ | $4.0\,\%$ |
| $\alpha$ | $(-0.8 \pm 11.5) \cdot 10^{-3}$ | $-0.11\,\%$ | $(6.4 \pm 13.8) \cdot 10^{-3}$ | $0.94\,\%$ |

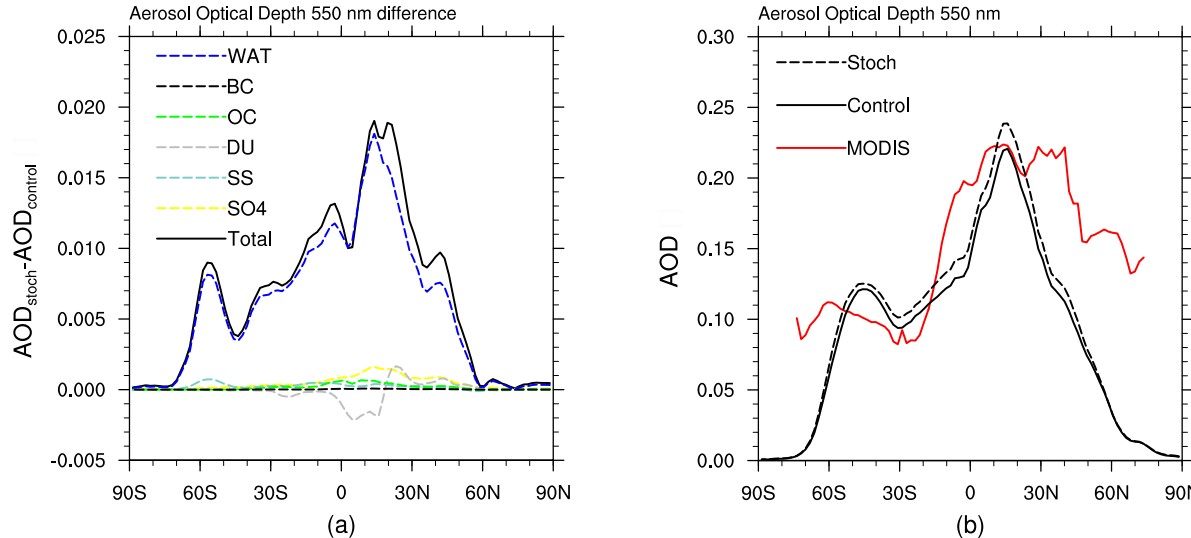

**Figure 3.** (a) Difference in AOD between the stochastic variation and the control run in AOD for different aerosol constituents. AOD from diagnosed aerosol water (WAT, blue) dominates the changes. (b) Temporally and zonally averaged clear-sky AOD from the control (plain) and stochastic (dashed) model runs (black, 01/2000 - 12/2009) and MODerate Resolution Imaging Spectroradiometer (Levy et al., 2013) Aqua satellite data (red, 08/2002 - 12/2010) is shown. Two discrepancies arise, namely (i) the fact that the model diagnoses clear-sky AOD also in overcast grid cells, with a relative humidity of RH=1 in these cases, which are dismissed in the MODIS retrieval, and (ii) the point that MODIS uses a conservative cloud masking, i.e. excludes pixels near cloud edges, whereas the model uses all clear-sky pixels.

on $RH_{cls}$ the expected value of $gf(RH_{cls})$ is greater than $gf(\overline{RH}_{cls})$ with $\overline{RH}_{cls}$ being the grid box mean clear-sky relative humidity.

Effects are stronger for aerosol particles with a larger radius, and thus, particles from CS and AS mode. Three reasons can explain the vertical shape of the $gf$-profiles:

5     (1) Clear-sky relative humidity has a decreasing trend with height in the model (see Fig. 2b). The same change $\Delta RH_{cls}$ in a drier environment leads to a smaller change in $gf$ than in a more humid environment (unless saturation is reached) because of the non-linearity of Eq. (1). The effect of the subgrid-scale variability of $RH_{cls}$ on $gf$ is therefore stronger in a more humid environment.

    (2) Very hygroscopic aerosol particles are more sensitive to changes in relative humidity and larger particles tend to become
10  deposited by impaction and sedimentation more easily. The main hygroscopic aerosol types are sulphate and sea salt, where sea salt ($\kappa = 1.12$) is more hygroscopic than sulphate ($\kappa = 0.6$). Sea salt particles are emitted at the surface of the ocean. Due to their high $\kappa$ value, sea salt particles grow strongly, are deposited easily and can not reach high altitudes. This is indicated in Fig. 4b that shows that the mixing ratio of sea salt decreases noticeably stronger with height than other aerosol compounds. Hence, the aerosol composition of the atmosphere shifts towards less hygroscopic components (smaller $\kappa$ values) with height and the
15  effect of perturbing relative humidity on hygroscopic growth becomes weaker with smaller $\kappa$ values. This is supported by the

study of Pringle et al. (2010) that examines the global distribution of $\kappa$ using the ECHAM/MESSy Atmospheric Chemistry (EMAC) model. They find that especially at marine sites $\kappa$-values decrease with height, whereas at continental sites $\kappa$ tends to be more constant with height.

(1) and (2) can only explain a decreasing trend of $\beta$ with height but not the maxima of the $\beta$ profiles between 600 and 700 hPa.

(3) The critical relative humidity determines the width of the PDF which is used to vary $RH_{cls}$ stochastically. The width $2\Delta RH_{cls}$ of the PDF is calculated by $\Delta RH_{cls} = 1 - RH_{crit}$ as described in section 2. But $RH_{crit}$ is a function of height (see Eq. (2)). It decreases from the surface to 600 hPa from 0.9 to close to 0.7. For higher altitudes, it is nearly constant and converges slowly towards 0.7 (see Fig. 2b). This in fact means that the width of the PDF increases with height from the surface to 600 hPa. Then, it is almost constant. The positive curvature of Eq. (1) implies that the wider the PDF is the stronger the mean hygroscopic growth. The increasing width of the PDF explains why $\beta$ becomes greater with height until 600 hPa. Above, effect (1) and (2) are dominant and $\beta$ decreases again.

The AOD, the effective extinction cross section, $\sigma$, and the ratio of scattering efficiency to total extinction efficiency, $\omega$, are enhanced because of the increase in the geometrical radius of the particles. Anthropogenic aerosols that arise in PD simulations are disproportionally hygroscopic. Therefore, hygroscopic aerosols swell stronger due to the new parameterization in PD than in PI simulations and scatter more solar radiation. This leads in turn to a higher AOD in PD than PI runs. The effect of the new parameterization is especially strong for lower latitudes because of the higher abundance of sea salt (not depicted) in these regions. In addition, anthropogenic emissions of sulphate are strong in China, India and over the Arab Peninsula and contribute to the peak of increased AOD in the northern tropics. Note, that ECHAM6-HAM2 currently does not simulate nitrate aerosols. The integration of nitrate aerosols will introduce very hygroscopic aerosols into the model that would alter our results. As Fig. 3b demonstrates, little can be said about improved skill of ECHAM-HAM2 to model AOD in respect to AOD satellite retrievals of MODIS-Aqua.

The net clear-sky solar radiation $SW_{net,cls}$ decreases ($\Delta SW_{net,cls} = -0.22\,\text{W m}^{-2}$) due to an increased reflection of solar radiation as indicated by an increased $\omega$. However, the effect on the net all-sky solar radiation $SW_{net}$ is greater ($\Delta SW_{net} = -0.34\,\text{W m}^{-2}$) than the effect on the net clear-sky solar radiation. This is maybe due to the fact that although total cloud cover ($TCC$) decreased by -0.08%, cloud cover is slightly enhanced in height levels between about 700 and 900 hPa (see graph for PD in Fig. 4). Hence, more solar radiation is reflected back to space by these clouds.

We proceed with the discussion of differences that arise between PD and PI simulations. The change in $SW_{net,cls}$ is stronger for PD than for PI emissions because backscattering of solar radiation is more enhanced by the new parameterization in PD than in PI simulations because anthropogenic aerosols are disproportionally hygroscopic. In addition, the response in RFari ($-0.04\,\text{W m}^{-2}$, $31\,\%$), indicates as well that the parameterization leads to stronger backscattering by aerosols. Unexpectedly, the response in $SW_{net}$ is greater in PI runs ($\Delta SW_{net,PI} = -0.47\,\text{W m}^{-2}$) than in PD runs ($\Delta SW_{net,PD} = -0.34\,\text{W m}^{-2}$). We assume that this might be due to the enhanced total cloud cover in the PI simulations ($\Delta TCC_{PI} = 0.17$) whereas total cloud cover decreased in PD simulations ($\Delta TCC_{PD} = -0.08$). We ascribe the differences in cloud cover to internal variability. Hence, we suspect that the converse results for $SW_{net}$ arise due to internal variability. The stronger increase in cloud cover for

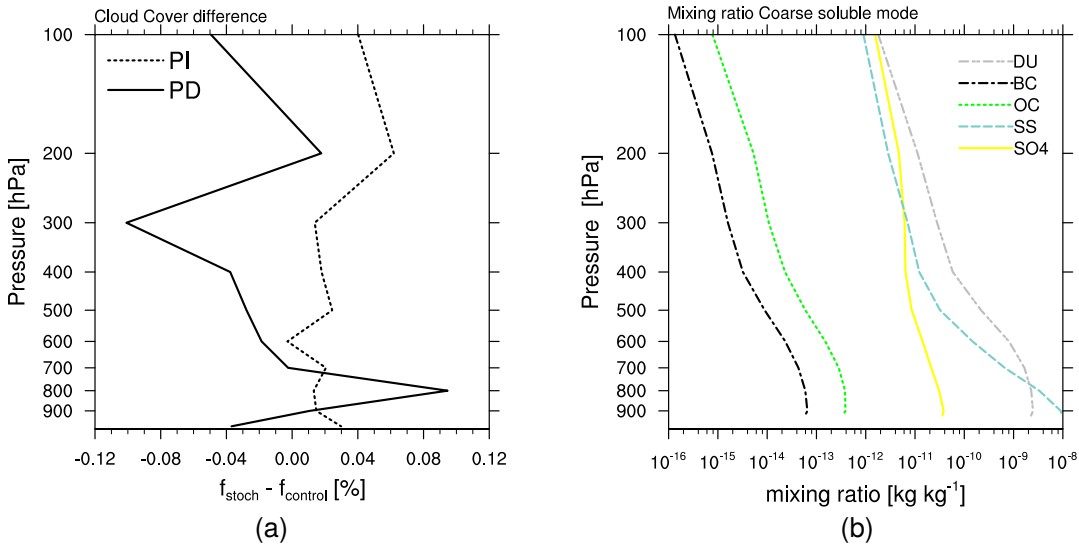

**Figure 4.** (a) Difference in cloud cover, $f$, due to the implementation of subgrid-scale variability of $RH_{cls}$ for PI (dashed) and PD (solid) simulations. (b) Global mean profile of mass mixing ratio for various aerosol compounds from the CS mode.

the higher troposphere in PI simulations (see Fig. 4) might explain the strong response $LW_{net}$ in PI simulations ($\Delta LW_{net,PI} = 0.26\,\mathrm{W\,m^{-2}}$). High-thin clouds, namely Cirrus clouds, are known to have positive effect on outgoing longwave radiation (Hartmann et al., 1992; Chen et al., 2000).

## 5 Conclusions

This study proposes a stochastic parameterization of clear-sky relative humidity that is consistent with the cloud cover scheme for its application in the aerosol hygroscopic growth parameterization. We investigate its effect on hygroscopic growth of aerosol particles as well as the subsequent changes of optical properties of the atmosphere and the radiative balance of the Earth. The implementation of the new parameterization leads to stronger swelling of aerosol particles (as expected) and therefore increases the AOD ($\sim 7.8\,\%$). Furthermore, the increased AOD causes stronger backscattering of solar radiation under clear-sky conditions $SW_{net,cls}$ ($-0.08\,\%$). Most importantly, the revision has a very strong influence on the simulated radiative forcing due to aerosol-radiation interaction RFari (31 %). In earlier studies RFari by sulphate increased in GCMs by about 10% when an idealized distribution for RH was implemented (Haywood and Shine, 1997; Haywood and Ramaswamy, 1998). Further studies found that GCMs underestimate RFari of sulphate when subgrid-scale variability of RH is not taken into account by 73 % in a limited-area-model case study (Haywood et al., 1997), by 30 to 80 % in a study that used a cloud-resolving model over a tropical ocean and a mid-latitude continental region (Petch, 2001) and by 30 to 40 % in a regional study (Europe and much of the North Atlantic) with a high resolution model (Myhre et al., 2002). Hence, our study is in line with previous studies

based on limited-area models. The effect of including RH subgrid variability is, however, bigger than the one found in the early global model study by Haywood and Ramaswamy (1998).

One might be able to further improve the parameterization of subgrid-scale variability of $RH_{cls}$ by applying the subsaturated part of the $\beta$-function from the optional Tompkins (2002) cloud cover scheme that prognostically treats the total-water variability PDF. Furthermore, Figure 2 in Quaas (2012) indicates that the critical relative humidity, $RH_{crit}$, that defines the width of the introduced $RH_{cls}$-PDF, varies horizontally in the same scale as vertically. Therefore, the width of the $RH_{cls}$-PDF could be extended from just height dependent to height and zonal or even height, zonal and meridional dependent.

## 6 Code availability

The code for the subgrid-scale variability of $RH_{cls}$ is available upon request from the first author.

## 7 Data availability

The ECHAM6-HAM2 model output data used in this study is archived at the Leipzig Institute for Meteorology and is available upon request from the authors. Satellite data from MODIS-Aqua can be obtained at https://neo.sci.gsfc.nasa.gov/. AEROCOM emission data can be downloaded at http://aerocom.met.no/emissions.html.

*Acknowledgements.* The ECHAM-HAM model is developed by a consortium composed of ETH Zurich, Max Planck Institut für Meteorologie, Forschungszentrum Jülich, University of Oxford, the Finnish Meteorological Institute and the Leibniz Institute for Tropospheric Research, and managed by the Center for Climate Systems Modeling (C2SM) at ETH Zurich. The MODIS data are from the NASA Goddard Space Flight Center (ftp://laadsweb.nascom.nasa.gov). The CERES data were obtained from the Atmospheric Sciences Data Center at NASA Langley Research Center (http://ceres.larc.nasa.gov). Funding by the European Research Council in Starting Grant, grant agreement FP7-306284 (QUAERERE) is acknowledged. We thank two anonymous reviewers for their work reviewing our manuscript.

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
