# Peer review of "Subgrid-scale variability of clear-sky relative humidity and forcing by aerosol-radiation interactions in an atmosphere model"

_Atmospheric Chemistry and Physics, 2017_

## Referee Comment (RC1) · S. J. Ghan (Referee) · 20 Nov 2017

This study introduces stochastic sampling of the PDF of humidity to examine subgrid humidification effects on aerosol radiative forcing. Although this represents an advance over previous estimates of aerosol radiative forcing, important details that could substantially influence the results are missing in the description of the treatment. I cannot recommend publication until these details are provided, and only then if the clarified treatment does not substantially bias the results.

1. Page 3, lines 23-28. How is the hygroscopicity of each mode determined from the hygroscopicity of each component in the modes?

[Figure]

2. Section 2.3 a. How is humidification effect on extinction treated? Extinction is not a simple function of particle radius. See, for example, the method of Ghan and Zaveri (2007). The treatment must be described and justified. b. Does the model treat absorption enhancement by humidification? Some people (Jacobson) think this is quite important. c. Why use the clear-sky value? This biases the estimate of ERFari. Why not include a diagnostic no-aerosol radiation calculation and diagnose ERFari following Ghan (2013)?

3. Page 7, last paragraph. Your argument about scattering vs absorption would be stronger if you compare the impact on AOD with the impact on AAOD. It is likely that the sensitivity of ERFari is biased by your treatment of humidification effects on absorption and by neglecting contributions from cloudy sky.

References

Ghan, S. J., and R. A. Zaveri, 2007: Parameterization of optical properties for hydrated internally-mixed aerosol. J. Geophys. Res., 112, D10201, doi:10.1029/2006JD007927.

Ghan, S. J., 2013: Technical note: Estimating aerosol effects on cloud radiative forcing. Atmos. Chem Phys., 13, 9971–9974, doi:10.5194/acp-13-9971-2013.

---

## Referee Comment (RC2) · Anonymous Referee #2 · 2 Dec 2017

This study applied a stochastic parameterization of subgrid-scale variability of relative humidity (RH) to a global climate-aerosol model, ECHAM6-HAM2, and examined the impact of the subgrid-scale variability of RH on aerosol optical depth (AOD) and radiative forcing. The authors showed the subgrid-scale variability of RH increased global mean aerosol hygroscopic growth, AOD (by 7.8%), and effective radiative forcing (by 57%) due to the non-linear response of hygroscopic growth to RH.

Although this study showed a slight improvement of the estimation of AOD, I don't think this study is suitable for a paper of Atmospheric Chemistry and Physics because the scientific findings, methods, and analysis of this study are not enough as shown below.

[Figure]

Main comments.

1) Page 2, lines 7-27

These two paragraphs describe about previous studies. I understand from these paragraphs that the underestimation of radiative forcing by using the grid-box mean RH is already recognized well in previous studies. In addition, there are some global model studies focused on the subgrid-scale variability of RH previously. Due to these two points, it is hard to understand what was advanced scientifically in this study.

This study is new in ECHAM6-HAM2, but I feel that there is no clear advancement both scientifically and technically in the community of aerosol and cloud studies.

2) Section 2.2

Please explain why the stochastic treatment was used. Does this mean only single RH value is calculated by Eq. (6) and used in each grid box and each time? I think considering the full range of RH (shown in Figure 1) in each grid box and time is not so difficult, for example by using 5-10 RH bins between RHcls - delta RHcls and RHcls + delta RHcls. This will not increase the computational cost of the model so much.

If a random RH is used in the model, does it assure the repeatability of model simulations? For example, when the authors make two simulations which use completely the same inputs and model setups, can the authors obtain the same results from the two simulations?

3) Treatment of aerosol absorption

How does the model calculate aerosol absorption? Please describe the method and the treatment of absorption enhancement by water. The treatment of absorption enhancement of black carbon by water will be a key in the calculations of single scattering albedo and radiative forcings. The authors show negative values of radiative forcings, but I suspect the authors do not consider the positive forcings by the absorption enhancement. The absolute values of radiative forcings will be smaller when the absorption enhancement is treated properly, and the total effect of the subgrid-scale variability of RH will be less important.

Other comments.

Page 1, lines 23-24:

I think Ginoux [2017] is for mineral dust only. References for primary and secondary organic aerosols and aerosols from biomass burning should be added.

Page 2, line 7:

"However" is better to move to the next sentence (before "General circulation").

Page 2, lines 34-35:

I cannot understand what the authors mean in this sentence ("RHcls is chosen...") and next sentence ("That means, ..."). I don't think "aerosol-radiation interactions are negligible in the cloud part". Some studies (e.g. Jacobson) have shown the importance of this issue.

Page 3, line 28:

I don't think "age due to internal mixing" is good explanation. Did the authors mean that internally-mixed particles are made by aging processes such as condensation and coagulation?

Page 4, line 15:

Because the authors used "usually" here, it looks there are some previous studies considering the subgrid-scale variability.

Page 4, Line 25:

The values of cs, ct, and nx should be given after the equation (2) (at line 8).

Page 5, line 2: What is "RHcls,old"?

Page 6, line 6:

"c" values (Equation (9)) are not useful in the current manuscript. Discuss more or remove from the manuscript.

Page 6, line 13:

How important the different growth factors between CS/AS and KS/NS modes is?

Page 6, line 18:

Why did AOD increase especially in the tropics?

Page 6, line 25:

Why don't you show the results of "c" by using a figure?

Page 6, line 29:

"w" means single scattering albedo?

Page 6, line 32:

The alpha change shown here is for wet particles? Please clarify. Please show the percentage of this change.

Page 7, line 14:

Did the authors show the definition of ERFaer? What is the difference between ERFaricls and ERFaer? The effect of total and anthropogenic aerosols is shown, respectively?

Page 7, line 15:

Why did cloud cover increase in PD simulations but decrease in PI simulations?

Page 9, line 13:

The authors focus on sulfate and sea salt here, but how about nitrate, ammonium, and secondary organic aerosol? How does the global model treat these aerosol species?

Page 10, lines 1-5:

Please show the simulation results obtained by the authors rather than citing previous studies.

Page 10, line 8:

Please explain why the function of height is used. The authors explain the treatment in the cloud scheme but do not explain whether the treatment is realistic.

Page 10, line 11:

"Eq. (Equation 1)" should be "Eq. (1)".

---

## Author Comment (AC1) · 11 Apr 2018

**Response to Steven Ghan**

We thank the reviewer for his constructive comments and for his thorough review. The reviewer comments are in plain font, the authors responses in *Italics*.

**General comments**

This study introduces stochastic sampling of the PDF of humidity to examine subgrid humidification effects on aerosol radiative forcing. Although this represents an advance over previous estimates of aerosol radiative forcing, important details that

could substantially influence the results are missing in the description of the treatment. I cannot recommend publication until these details are provided, and only then if the clarified treatment does not substantially bias the results.

*The reviewers concerns are to our understanding mostly based on our insufficient description of the aerosol-module HAM2. Therefore, we include a new subsection, "2.1 The aerosol module HAM2" into our methods that briefly summarizes the properties of HAM2.*

**1. Page 3, lines 23-28.** How is the hygroscopicity of each mode determined from the hygroscopicity of each component in the modes?

*In ECHAM6-HAM2 the hygroscopicity of internally-mixed aerosols is determined by calculating the volume weighted sum of the $\kappa$-values form each soluble compound (see Zhang et al. (2012) section 4.1.3). This is now better explained in the revised paper.*

**2. Section 2.3 a.** How is humidification effect on extinction treated? Extinction is not a simple function of particle radius. See, for example, the method of Ghan and Zaveri (2007). The treatment must be described and justified.

*Aerosol radiative properties are calculated using Mie theory. The model uses volume-averaging for each of the seven aerosol modes to calculate the refractive indices where aerosol water is included using the ambient relative humidity. The effective complex radiative indices and the Mie size parameter is then used for the aerosol radiative properties, namely extinction cross section, single scattering albedo, and asymmetry parameter in the radiation scheme (see Zhang et al., 2012, section 2.6).*

*We thank Dr Ghan for pointing us to this reference, which we now include in the discussion in the revised paper.*

**b.** Does the model treat absorption enhancement by humidification? Some people (Jacobson) think this is quite important.

*For the version 2 of the aerosol module HAM the refractive indices for black carbon were updated to reduce the the negative biased aerosol absorption enhancement (Stier et al., 2007). In Stier et al. (2007) it is argued that on a global scale the absorption enhancement of BC due to mixing with hydrophilic aerosols is compensated by the lower life time of and abundance of BC. They base this argument on the study by Stier et al. (2006) where they find that reduced lifetime of BC due to internal mixing actually overbalances the absorption enhancement effect on a global scale such that they observe a decrease in global annual mean clear-sky atmospheric absorption of $0.2\,W\,m^{-2}$. Furthermore, the hypothesis of Jacobson (2012) is very controversial and not supported by most other studies (e.g. Twohy et al., 1989; Chýlek et al., 1996; Liu et al., 2002).*

**c.** Why use the clear-sky value? This biases the estimate of ERFari. Why not include a diagnostic no-aerosol radiation calculation and diagnose ERFari following Ghan (2013)?

*We followed the advice of the referee. We now present our results in terms of a radiative forcing due to aerosol-radiation interactions (RFari) that is calculated as suggested by Ghan (2013). As one would expect, RFari changes less (from -0.15 to -0.19 $W\,m^{-2}$, $\sim$31%) due to the masking effect of clouds than ERFari$_{cls}$ (from -0.29 to -0.45 $W\,m^{-2}$, $\sim$57%) in response to the implementation of our parameterization. However, we still*

*observe a clear signal that the implementation of our parameterization enhances the cooling by the direct aerosol effect.*

**3. Page 7, last paragraph.** Your argument about scattering vs absorption would be stronger if you compare the impact on AOD with the impact on AAOD. It is likely that the sensitivity of ERFari is biased by your treatment of humidification effects on absorption and by neglecting contributions from cloudy sky.

*We have included two variables into the Table 1, namely AAOD and AAOD by black carbon. We see a positive change in AAOD +0.12 $\cdot$ $10^{-3}$ ($\sim$ 4.7%). However, AOD changed in absolute values by nearly two orders of magnitude more, namely + 9.0 $\cdot$ $10^{-3}$ ($\sim$ 7.8%).*

**References**

Chýlek, P., Lesins, G. B., Videen, G., Wong, J. G. D., Pinnick, R. G., Ngo, D., and Klett, J. D.: Black carbon and absorption of solar radiation by clouds, J. Geophys. Res., 101, 23 365–23 371, doi:10.1029/96JD01901, 1996.

Ghan, S. J.: Technical Note: Estimating aerosol effects on cloud radiative forcing, Atmos. Chem. Phys., 13, 9971–9974, doi:10.5194/acp-13- 9971-2013, 2013.

Ghan, S. J. and Zaveri, R. A.: Parameterization of optical properties for hydrated internally mixed aerosol, J. Geophys. Res., 112, D10 201, doi:10.1029/2006JD007927, 2007.

Jacobson, M. Z.: Investigating cloud absorption effects: Global absorption properties

of black carbon, tar balls, and soil dust in clouds and aerosols, J. Geophys. Res., 117, D06 205, doi:10.1029/2011JD017218, 2012.

Liu, L., Mishchenko, M. I., Menon, S., Macke, A., and Lacis, A. A.: The effect of black carbon on scattering and absorption of solar radiation by cloud droplets, Journal of Quantitative Spectroscopy and Radiative Transfer, 74, 195–204, doi:10.1016/S0022-4073(01)00232-1, 2002.

Stier, P., Seinfeld, J. H., Kinne, S., Feichter, J., and Boucher, O.: Impact of nonabsorbing anthropogenic aerosols on clear-sky atmospheric absorption, J. Geophys. Res., 111, D18 201, doi:10.1029/2006JD007147, 2006.

Stier, P., Seinfeld, J. H., Kinne, S., and Boucher, O.: Aerosol absorption and radiative forcing, Atmospheric Chemistry and Physics, 7, 5237–5261, 2007.

Twohy, C. H., Clarke, A. D., Warren, S. G., Radke, L. F., and Charlson, R. J.: Light-absorbing material extracted from cloud droplets and its effect on cloud albedo, J. Geophys. Res., 94, 8623–8631, doi:10.1029/JD094iD06p08623, 1989.

Zhang, K., O'Donnell, D., Kazil, J., Stier, P., Kinne, S., Lohmann, U., Ferrachat, S., Croft, B., Quaas, J., Wan, H., Rast, S., and Feichter, J.: The global aerosol-climate model ECHAM-HAM, version 2: sensitivity to improvements in process representations, Atmos. Chem. Phys., 12, 8911–8949, doi:10.5194/acp-12-8911-2012, 2012.

---

## Author Response (AR1)

**Response to Steven Ghan**

We thank the reviewer for his constructive comments and for his thorough review. The reviewer comments are in plain font, the authors responses in *Italics*.

**General comments**

This study introduces stochastic sampling of the PDF of humidity to examine subgrid humidification effects on aerosol radiative forcing. Although this represents an advance over previous estimates of aerosol radiative forcing, important details that

could substantially influence the results are missing in the description of the treatment. I cannot recommend publication until these details are provided, and only then if the clarified treatment does not substantially bias the results.

*The reviewers concerns are to our understanding mostly based on our insufficient description of the aerosol-module HAM2. Therefore, we include a new subsection, "2.1 The aerosol module HAM2" into our methods that briefly summarizes the properties of HAM2.*

**1. Page 3, lines 23-28.** How is the hygroscopicity of each mode determined from the hygroscopicity of each component in the modes?

*In ECHAM6-HAM2 the hygroscopicity of internally-mixed aerosols is determined by calculating the volume weighted sum of the $\kappa$-values form each soluble compound (see Zhang et al. (2012) section 4.1.3). This is now better explained in the revised paper.*

**2. Section 2.3 a.** How is humidification effect on extinction treated? Extinction is not a simple function of particle radius. See, for example, the method of Ghan and Zaveri (2007). The treatment must be described and justified.

*Aerosol radiative properties are calculated using Mie theory. The model uses volume-averaging for each of the seven aerosol modes to calculate the refractive indices where aerosol water is included using the ambient relative humidity. The effective complex radiative indices and the Mie size parameter is then used for the aerosol radiative properties, namely extinction cross section, single scattering albedo, and asymmetry parameter in the radiation scheme (see Zhang et al., 2012, section 2.6).*

[Figure]

*We thank Dr Ghan for pointing us to this reference, which we now include in the discussion in the revised paper.*

**b.** Does the model treat absorption enhancement by humidification? Some people (Jacobson) think this is quite important.

*For the version 2 of the aerosol module HAM the refractive indices for black carbon were updated to reduce the the negative biased aerosol absorption enhancement (Stier et al., 2007). In Stier et al. (2007) it is argued that on a global scale the absorption enhancement of BC due to mixing with hydrophilic aerosols is compensated by the lower life time of and abundance of BC. They base this argument on the study by Stier et al. (2006) where they find that reduced lifetime of BC due to internal mixing actually overbalances the absorption enhancement effect on a global scale such that they observe a decrease in global annual mean clear-sky atmospheric absorption of $0.2\,W\,m^{-2}$. Furthermore, the hypothesis of Jacobson (2012) is very controversial and not supported by most other studies (e.g. Twohy et al., 1989; Chýlek et al., 1996; Liu et al., 2002).*

**c.** Why use the clear-sky value? This biases the estimate of ERFari. Why not include a diagnostic no-aerosol radiation calculation and diagnose ERFari following Ghan (2013)?

*We followed the advice of the referee. We now present our results in terms of a radiative forcing due to aerosol-radiation interactions (RFari) that is calculated as suggested by Ghan (2013). As one would expect, RFari changes less (from -0.15 to -0.19 $W\,m^{-2}$, $\sim$31%) due to the masking effect of clouds than ERFari$_{cls}$ (from -0.29 to -0.45 $W\,m^{-2}$, $\sim$57%) in response to the implementation of our parameterization. However, we still*

[Figure]

*observe a clear signal that the implementation of our parameterization enhances the cooling by the direct aerosol effect.*

**3. Page 7, last paragraph.** Your argument about scattering vs absorption would be stronger if you compare the impact on AOD with the impact on AAOD. It is likely that the sensitivity of ERFari is biased by your treatment of humidification effects on absorption and by neglecting contributions from cloudy sky.

*We have included two variables into the Table 1, namely AAOD and AAOD by black carbon. We see a positive change in AAOD +0.12 $\cdot$ $10^{-3}$ ($\sim$ 4.7%). However, AOD changed in absolute values by nearly two orders of magnitude more, namely + 9.0 $\cdot$ $10^{-3}$ ($\sim$ 7.8%).*

[Figure]

We thank the reviewer for his/her constructive comments and for his/her thorough review. The reviewer comments are in plain font, the responses in *Italics*.

**General comments**

This study applied a stochastic parameterization of subgrid-scale variability of relative humidity (RH) to a global climate-aerosol model, ECHAM6-HAM2, and examined the impact of the subgrid-scale variability of RH on aerosol optical depth (AOD) and

radiative forcing. The authors showed the subgrid-scale variability of RH increased global mean aerosol hygroscopic growth, AOD (by 7.8%), and effective radiative forcing (by 57%) due to the non-linear response of hygroscopic growth to RH. Although this study showed a slight improvement of the estimation of AOD, I don't think this study is suitable for a paper of Atmospheric Chemistry and Physics because the scientific findings, methods, and analysis of this study are not enough as shown below.

*To emphasize the scientific importance of applying a stochastic parameterization to clear-sky relative humidity for its application in the aerosol hygroscopic growth scheme we now highlight better in the revised manuscript that our study is the first study that is proposed without strong simplifications about the shape of the used probability density function (PDF) and that is consistent with the cloud cover scheme. This is discussed in more detail further down in this authors response.*
*Furthermore, we also emphasize in the revision the point that applying a stochastic parameterization is not only a method to estimate uncertainties but leads to a better representation of the mean state of the atmosphere. This was recently summarized in Berner et al. (2017). To highlight this in an example we refer to Tompkins and Berner (2008) that use a method of subgrid-scale variability that is very similar to ours. They investigate its influence when it is applied on the convective scheme of the European Centre for Medium-Range Weather Forecasts (ECMWF) ensemble prediction system. They show that their new stochastic convective scheme generally improves the skill of the operational system for most variables in the short to medium range in mid-latitudes.*

**Main comments**

**1) Page 2, lines 7-27**
These two paragraphs describe about previous studies. I understand from these paragraphs that the underestimation of radiative forcing by using the grid-box mean RH is

already recognized well in previous studies. In addition, there are some global model studies focused on the subgrid-scale variability of RH previously. Due to these two points, it is hard to understand what was advanced scientifically in this study. This study is new in ECHAM6-HAM2, but I feel that there is no clear advancement both scientifically and technically in the community of aerosol and cloud studies.

*We gather that the reviewer refers to our discussion of the studies of Haywood and Shine (1997) and Haywood and Ramaswamy (1998) which apply a subgrid-scale variability of RH in a GCM. We regret that in the previous manuscript version obviously we did not clarify well enough that our study goes substantially beyond this previous work.*

*1. Haywood and Shine (1997) investigated the effect of subgrid-scale variability in an idealized case. They use globally for each grid cell and height level five fixed RH-values that are taken from a normal distribution around RH = 70%. Hence, they show the gross effect of subgrid-scale variability of RH but do not propose a scheme that is meant to be integrated into an atmosphere model.*

*2. Haywood and Ramaswamy (1998) use a more sophisticated approach by computing the subgrid-scale variability based on a triangular shaped distribution around grid-box mean RH. However, the shape and width of the distribution is globally constant. In our parameterization the width of the PDF is a function of height (Quaas, 2012). Furthermore, the PDF that Haywood and Ramaswamy implement is artificially generated and inconsistent with the assumptions in the cloud scheme. In contrast, we sample the sub-saturated part of the PDF from the cloud-cover scheme.*

*Haywood and Shine (1997) and Haywood and Ramaswamy (1998) have in common that they just investigate the effects on RFari by sulphate. We investigate the effect on the entire radiative forcing of aerosols that are included in the ECHAM6-HAM2 model. Summarizing, our study is the first study that investigated the effect of subgrid-scale variability in an approach that does not make idealized assumptions and that is consistent with the cloud cover scheme.*

[Figure]

*In response to the reviewer comment, the sentence: "Haywood and Shine (1997) and Haywood and Ramaswamy (1998) include a subgrid-scale variability of RH for the calculation of RFari by sulphate."*
*in lines 17f of the original manuscript has been amended as follows:*
*"First attempts to implement a subgrid-scale variability of RH in a GCM for the calculation of RFari by sulphate were made by Haywood and Shine (1997) and Haywood and Ramaswamy (1998). However, these studies make strong simplifications about the shape of the used probability density function (PDF) and are not consistent with the cloud cover scheme."*

**2) Section 2.2**

Please explain why the stochastic treatment was used. Does this mean only single RH value is calculated by Eq. (6) and used in each grid box and each time? I think considering the full range of RH (shown in Figure 1) in each grid box and time is not so difficult, for example by using 5-10 RH bins between RHcls - delta RHcls and RHcls + delta RHcls. This will not increase the computational cost of the model so much. If a random RH is used in the model, does it assure the repeatability of model simulations? For example, when the authors make two simulations which use completely the same inputs and model setups, can the authors obtain the same results from the two simulations?

*At each time step and each grid box we apply Eq. (6) using a newly generated random number. This is computationally cheaper than subsampling the entire PDF in each time step while on average one expects a very similar result. It should be noted that our scheme is intended for use in a 3-D climate model and not just constructed to show that taking into account subgrid scale variability decreases ERFari. Applying a binning approach for the RH values means, in other words, increasing the resolution of the model for the hygroscopic growth scheme. This results in additional computational*

[Figure]

*cost compared with the approach suggested here.*
*Up until now, we used a random number generator that starts always with a new seed. In general it would be possible to start always with the same seed. The repeatability of our study is ensured by integrating the model over a rather long time (10 years). Hence, we expect to find results that do not differ from our results larger than within the given error bars (95%). In the current setting, the integration is not fully deterministic anymore. We clarify this now in the revised text.*

**3) Treatment of aerosol absorption**
How does the model calculate aerosol absorption? Please describe the method and the treatment of absorption enhancement by water. The treatment of absorption enhancement of black carbon by water will be a key in the calculations of single scattering albedo and radiative forcings. The authors show negative values of radiative forcings, but I suspect the authors do not consider the positive forcings by the absorption enhancement. The absolute values of radiative forcings will be smaller when the absorption enhancement is treated properly, and the total effect of the subgrid-scale variability of RH will be less important.

*Aerosol radiative properties are calculated using Mie theory. The model uses volume-averaging for each of the seven aerosol modes to calculate the refractive indices where aerosol water is included using the ambient relative humidity. The effective complex radiative indices and the Mie size parameter is then used for the aerosol radiative properties, namely extinction cross section, single scattering albedo, and asymmetry parameter in the radiation scheme (see Zhang et al. (2012) section 2.6). The explanation is now extended in the revised manuscript.*
*For the version 2 of the aerosol module HAM the refractive indices for black carbon were updated to reduce the negative bias aerosol absorption enhancement (Stier et al., 2007). Based on findings of Stier et al. (2006) it is argued in Stier et al. (2007)*

[Figure]

*that the absorption enhancement of BC due to mixing with hydrophilic aerosols is compensated by the lower life time of and abundance of BC.*

*In contrast to Jacobson (2012) and Bond et al. (2013), HAM2 does not include a very strong absorption enhancement for absorbing particles inside clouds. This is because the hypothesis of Jacobson (2012) is very controversial and not supported by most other studies (e.g. Twohy et al., 1989; Chýlek et al., 1996; Liu et al., 2002).*

*We include values for the AAOD and AAOD of BC. In PD simulations AAOD increases by +0.12 ($\pm$0.4) $\cdot$ $10^{-3}$ ($\sim$ 4.7%) were the AAOD by BC increased by +0.11 ($\pm$0.3) $\cdot$ $10^{-3}$ ($\sim$5.1%). This shows that our parameterization leads to a stronger absorption of solar light by BC aerosols. However, the overall increase of AOD in PD simulations due to our new parameterization is +9.0 ($\pm$2.2) $\cdot$ $10^{-3}$ ($\sim$7.8%). That highlights that in our simulations the contribution of absorption to AOD is rather low although absorption is enhanced.*

*For point 3, the reviewers concerns are to our understanding mostly based on our insufficient description of the aerosol-module HAM2. Therefore, we include a new subsection, "2.1 The aerosol module HAM2" into our methods that briefly summarizes the properties of HAM2.*

**Other comments**

**Page 1, lines 23-24:**
I think Ginoux [2017] is for mineral dust only. References for primary and secondary organic aerosols and aerosols from biomass burning should be added.

*We included Bond et al.(2013) and Myhre et al. (2013) for biomass burning, Shindell et al. (2013) for SOA as well as a reference to the AR5 from the IPCC.*

[Figure]

**Page 2, line 7:**
"However" is better to move to the next sentence (before "General circulation").

*Changed.*

**Page 2, lines 34-35:**
I cannot understand what the authors mean in this sentence ("RHcls is chosen. . .")
and next sentence ("That means, . . ."). I don't think "aerosol-radiation interactions are
negligible in the cloud part". Some studies (e.g. Jacobson) have shown the importance
of this issue.

*In the standard configuration ECHAM6-HAM2 uses the mean clear-sky relative humid-
ity ($\overline{RH}_{\text{cls}}$) instead of the grid-box mean relative humidity ($\overline{RH}$) to compute aerosol
hygroscopic growth. Note, that $\overline{RH}_{\text{cls}}$ is by definition smaller than $\overline{RH}$:*

$$\overline{RH} = f \cdot \overline{RH}_{\text{cloud}} + (1 - f) \cdot \overline{RH}_{\text{cls}} = f + (1 - f) \cdot \overline{RH}_{\text{cls}}$$

*Here, $f$ is the fractional cloud cover. It is assumed that the radiative effect of the
hygroscopic growth of aerosol is more important in the cloud-free part than for
interstitial aerosol in clouds where cloud radiative effects are dominant. If not RH$_{\text{cls}}$
but $\overline{RH}$ would be used to calculate the aerosol hygroscopic growth for the entire
grid-box, aerosols would grow to strong in the cloud free part where aerosol-radiation
interactions are of higher importance than in the cloudy sky. Thus, it is reasoned in the
ECHAM literature to use RH$_{\text{cls}}$ instead $\overline{RH}$ (see section 2.6 in Stier et al., 2005).
We added to the method section:
"RH$_{\text{cls}}$ is chosen, instead of grid-box mean relative humidity RH, because cloud
processing and cloud radiative effects are dominant in the cloudy part of a grid box as
reasoned in Stier et al. (2005) for ECHAM5-HAM1."*

[Figure]

**Page 3, line 28:**
I don't think "age due to internal mixing" is good explanation. Did the authors mean that internally-mixed particles are made by aging processes such as condensation and coagulation?

*Yes, that is what we meant. We changed the sentence to:*
*"DU and BC are considered as non-hygroscopic on emission. However, they can merge with hygroscopic particles due to internal mixing by ageing processes such as condensation and coagulation."*

**Page 4, line 15:**
Because the authors used "usually" here, it looks there are some previous studies considering the subgrid-scale variability.

*We just want to highlight that the idea of subgrid-scale variability is not new to model formulations (e.g. vertical velocity) but was not applied (besides in idealized studies as mentioned further up) in global circulation models to RH or RH$_{cls}$. We deleted the "usually" changed the sentence to:*
*"Several global atmosphere models including ECHAM6-HAM2 already make assumptions to account for the subgrid-scale variability of atmospheric variables, e.g. for vertical velocity when computing droplet activation rates (Ghan et al., 1997; Lohmann et al., 2007; Golaz et al., 2011). However, subgrid-scale variability of RH or RH$_{cls}$ is not taken into account when computing hygroscopic growth of interstitial aerosols besides in some studies that made gross simplification regarding the shape and variation of the used PDF (Haywood and Shine, 1997; Haywood and Ramaswamy 1998)."*

**Page 4, Line 25:**

[Figure]

The values of cs, ct, and nx should be given after the equation (2) (at line 8).

*Changed.*

**Page 5, line 2:**
What is "Rhcls,old"?

*We changed the expression to $\overline{RH}_{cls}$. This variable is used earlier to indicate the grid-box mean clear-sky relative humidity that was actually meant by $RH_{cls,old}$.*

**Page 6, line 6:**
"c" values (Equation (9)) are not useful in the current manuscript. Discuss more or remove from the manuscript.

*We agree with the referee and removed them from the manuscript.*

**Page 6, line 13:**
How important the different growth factors between CS/AS and KS/NS modes is?

*In Figure 2a one can see that particles swell stronger for bigger aerosol modes due to the implementation of the new parameterization. For more clarity, we changed the sentence:*
*"Thus, the effect is stronger for particles from CS and AS mode than for particles from KS and NS mode. "*
*to:*
*"Thus, the effect is strongest for particles from the CS mode (red line in Fig. 2a) and*

[Figure]

*weakest for particles from the NS mode (black line in Fig. 2a)"*

**Page 6, line 18:**
Why did AOD increase especially in the tropics?

*Our parameterization leads to an on average stronger growth of hygroscopic aerosols, especially for very hygroscopic aerosols that are sulphate and sea salt.*
*Sea salt is most abundant in lower latitudes in the model. This is indicated in Figure 1 that we attached to this response. Depicted is the AOD by non-hydrated sea salt aerosols. Furthermore, anthropogenic emissions of sulphate are very high in China, India and over the Arab Peninsula and contribute in addition to the peak of increased AOD in the northern tropics.*

**Page 6, line 25:**
Why don't you show the results of "c" by using a figure?

*We excluded the $c$-value completely from the paper in response to the previous reviewer comment on "Page 6, line 6".*

**Page 6, line 29:**
"w" means single scattering albedo?

*Yes. But we use term "ratio of scattering to total extinction" when we refer to results from the radiation transfer equation to compute the ratio between scattering/extinction. We do this in order to make the mentioned variable clearly distinguishable from the single scattering albedo as a property of a certain particle type (that is constant).*

[Figure]

**Page 6, line 32:**
The alpha change shown here is for wet particles? Please clarify. Please show the percentage of the change.

*Yes it is the parameter for wet particles. We added this and the percentage of this change.*

**Page 7, line 14:**
Did the authors show the definition of ERFaer?

*We added a description of how ERFaer was computed.*

What is the difference between ERFari-cls and ERFaer?

*In response to the comments of Steven Ghan, we now discuss our results in terms of radiative forcing due to aerosol-radiation interactions (RFari) to avoid confusion due to the earlier used ERFari$_{cls}$, that was a idealized value that depicted the radiative inbalance at the top of atmosphere in a hypothetical atmosphere without clouds.*

The effect of total and anthropogenic aerosols is shown, respectively?

*We just show the ERF due to anthropogenic aerosols (ERFaer) since we compute the difference between PD and PI emissions. We now mention this in the methods section in the manuscript.*

[Figure]

**Page 7, line 15:**
Why did cloud cover increase in PD simulations but decrease in PI simulations?

*We ascribe this to internal variability. That can be seen by overlapping confidence intervals:*
TCC$_{PD}$: -0.08 $\pm$ 0.14%
TCC$_{PI}$: 0.17 $\pm$ 0.14%

*We added a sentence about internal variability to the discussion.*

**Page 9, line 13:**
The authors focus on sulfate and sea salt here, but how about nitrate, ammonium, and secondary organic aerosol? How does the global model treat these aerosol species?

*We focus on sulphate and sea salt because they are very hygroscopic (for HAM2 $\kappa_{SS}$ = 1.12 and $\kappa_{SO4}$ = 0.6, see Zhang et al., 2012) in comparison to SOA ($\kappa_{SOA}$ = 0.037). Aerosols that are more hygroscopic are more sensitive to changes in relative humidity than aerosols that are less hygroscopic. This characteristic is what we use for the explanation of the observed profile of the change in the growth factor. Note, that the $\kappa$-value for sulphate in HAM2 is in the range of the observed value for ammonium sulphate (0.33 - 0.72) (Petters and Kreidenweise, 2007). Furthermore, the model that we use currently does not simulate nitrate aerosols (Stier et al., 2005). We highlight in our paper now that the addition of nitrate aerosols will introduce very hygroscopic aerosols into the model that would alter our results.*

**Page 10, lines 1-5:**
Please show the simulation results obtained by the authors rather than citing previous

studies.

*We now integrate a profile plot that shows that the mixing ratio of hygroscopic aerosols. The plot shows that the mixing ratio of sea salt decreases stronger with height than for other aerosols. Hence, the overall aerosols composition becomes less hygroscopic.*

**Page 10, line 8:**
Please explain why the function of height is used. The authors explain the treatment in the cloud scheme but do not explain whether the treatment is realistic.

*$RH_{crit}$ is a function of height in the general formulation of ECHAM6-HAM2. Implicitly we already stated that in the introduction when we referred to Quaas (2012). However, now clearly formulate that Quaas (2012) found $RH_{crit}$ to be a function of height.*

**Page 10, line 11:**
"Eq. (Equation 1)" should be "Eq. (1)".

*Changed.*

**References**

[revised manuscript text omitted]

**Page 1, line 12f**
Results are now presented in terms of RFari instead of ERFari_cls.

**Page 1, line 24**
We included literature for the radiative forcing by different aerosol compounds.

**Page 2, line 14ff**
We moved the following sentence to the end of the paragraph.

"In addition, recent studies show that models with a coarse resolution which do not take subgrid-scale variability of various aerosol properties into account underestimate aerosol radiative forcing (Gustafson et al., 2011) and have a significant negative bias in aerosol optical depth (Weigum et al., 2016) ."

**Page 2, line 19ff**
We included a more detailed description about Haywood and Shine (1997) and Haywood and Ramaswamy (1998)  to show that our study goes substantially beyond this previous
work.

**Page 3, line 16ff**
We included a paragraph about Tompkins and Berner (2008) to show a successful application of  a method of subgrid-scale variability of humidity that is very similar to ours.

**Page 3, line 27ff**
We included an entire subsection about the aerosol module HAM2. We now explain in more detail how HAM2 calculates refractive indices and treats absorption enhancement of BC.

**Page 4, line 23ff**
We included statements about the hygroscopicity of nitrate, ammonium and internally-mixed aerosols in HAM2.

**Page 4, line 30ff**
We explain in more detail why HAM2 uses in its standard setup RH_cls instead RH in the hygroscopic growth scheme.

**Page 4, Table 1**
We included a table for the kappa values as they are used in HAM2.

**Page 5, line 8ff**
The paragraph about aerosol modes is moved to the new subsection about the module HAM2.

**Page 7, line 3f**
We included a sentence about the repeatability of our study.

**Page 7, line 10ff**
We included the definition of the total ERF by anthropogenic aerosols, ERFaer, that we use, excluded the definition of ERFari_cls and included the definition of RFari.

**Page 8, line 12ff**
We excluded the discussion of our model results in terms of the variable "c" from the entire manuscript.

**Page 8, line 19ff**
The statement about the confidence interval is moved from page 8, line 28f to page 8, line 19ff.

**Page 9, line 3ff**
Again, we excluded the discussion of our model results in terms of the variable "c".

**Page 9, line 8ff**
We included results for the absorption aerosol optical depth.

**Page 9, line 16ff**
We moved the results regarding cloud cover from page 9, line 35ff to page 9, line 16ff.

**Page 9, line 29f**
Results are now presented in terms of RFari instead if ERFari_cls.

**Page 11, Table 2**
We included results for RFari, AOD_WAT, AAOD and AAOD_BC. We removed the results for ERFari_cls.

**Page 12, line 8f**
We included a statement about the profile of the mixing ratio of soluble aerosol compounds to underline our argument made in point (2).

**Page 13, line 13ff**
We included an explanation why the effect of the new parameterization is especially strong for lower latitudes.

**Page 14, Figure 1b**
We included a figure depicting the profile of the mixing ratio of various aerosol compounds.

**Page 14, line 1f**
We included a sentence regarding the internal variability of cloud cover in our model.

**Page 14, line 9f**
We now briefly highlight the advances of our study in respect to previous studies in the conclusions.

**Page 14, line 5f**
We excluded the comparison of our results to data from MODIS-Aqua.

[revised manuscript text omitted]

---

## Author Response (AR2)

**Authors Response**

Dear Editor,

Thank you very much for editing our manuscript. Please find a revised version of our manuscript and our response to the referee report attached.

We have revised the manuscript in order to clarify why we believe that this study is suited for ACP. In particular we now highlight the fact that this is the first time that the influence of subgrid scale RH variability on RFari is computed in a global model with a parameterization for sub-grid variability of RH that is consistent with the assumptions in the model already in the abstract.

However, in case this is not sufficient to address your concerns, we would not object if you decided to publish it as a "technical note" in ACP.

Kind regards
Paul

**Reply to Referee #2**:
We thank the referee for a number of very constructive comments. In the following the referee's comments are in **bold font** and the author's responses in plain font.

**The authors revised the manuscript substantially following the comments/suggestions by the two reviewers. The manuscript was much improved in this revision. However, I still think this is a technical manuscript and more suitable for GMD or a technical note in ACP.**

The main purpose of this manuscript is to provide a new estimate of the influence of subgrid scale variability of RH on RFari. In our opinion this is primarily a scientific goal. In order to achieve this goal, we have used a state-of-the art 3-D circulation model in which we have implemented a consistent treatment of RH subgrid scale variability. We estimate that the effect is about three times as large as the only other estimate that is based on a global model. In order to clarify this, we changed the sentence

"Hence, our study is in accordance with previous investigations."

in the conclusion section to:

"Hence, our study is in line with previous studies based on limited area models. The effect of including RH subgrid variability is, however, bigger than the one found in the early global model study by Haywood and Ramaswamy (1998)."

In order to highlight that this is the first time that the effect of subgrid variability on RFari is computed using a global model we have included the following sentence in the abstract:

"While previous studies based on limited-area and global models suggest that sub-grid scale variability of RH should be taken into account for estimating RFari, here we present the first estimate of RFari using a global atmospheric model with a parameterization for sub-grid scale variability of RH that is consistent with the assumptions in the model."

Please also refer to our next reply below. We also clarified that Haywood et al. (1997); Petch (2001); Myhre et al. (2002) used limited area models in their studies. Only Haywood and Ramaswamy (1998) used a global model.

**Comment on reply to major comment 1: A main reason why I feel this is a technical manuscript is that it is not clear whether the improvement of the sub-grid treatment by the authors (e.g., stochastic treatment, consistency with the cloud scheme) is really scientifically important or not. The authors describe that previous studies make strong simplifications about the shape of the used probability density function (PDF) and are not consistent with the cloud cover scheme" (page 2, lines 16-17) and that they do not consider variations of width and shape of the used distribution." (page 2, lines 26-27). The authors also describe that the width of the PDF is a function of height and the sub-saturated part of the PDF is sampled from the cloud-cover scheme in the parameterization developed in this study. Please show and explain how important the improvement by the authors is in the RF estimation (compared with the previous studies).**

We agree that it is indeed a shortcoming of this study that we did not also test the assumptions used by Haywood and Ramaswamy (1998) in our model. Thus, it is not clear why exactly our estimate differs from the earlier estimate. However, we also feel that doing so would be outside the scope of this study. Haywood and Ramaswamy (1998) solely applied a triangular PDF of with a width of $\pm 10\%$ around the mean RH. Such a configuration can be regarded as a choice without any strong justification by observations. In contrast, the variation of the width of our PDF is consistent with the cloud cover scheme and is supported by observations (Quaas, 2012). Furthermore, Haywood and Ramaswamy (1998) only consider sulphate, while we use a subgrid-scale parameterization of RH in a "state-of-the art" aerosol-climate model (which, however, does not include "absorption enhancement").

In order to highlight the differences between our study and the study by Haywood and Ramaswamy (1998), we added the following sentence after the sentence "do not consider variations of width and shape of the used distribution." (page 2, lines 26-27):

"This is a rather strong simplification (especially having the non-linear hygroscopic growth in mind) in comparison to findings of Quaas (2012) who suggests a change of the width of a uniform PDF from about $\pm 20\%$ at the surface to about $\pm 65\%$ in the middle troposphere."

**I understand that 1) the importance of resolving sub-grid RH variability itself was already shown in the previous studies and 2) the treatment of sub-grid RH variability was updated in this study. Since the authors focus on the comparisons between the simulation with the stochastic parameterization and the simulation without the sub-grid RH variability, both effects (1 and 2) are included in the author's comparisons. I think the importance of the second effect should be shown and explained if the authors want to show the improvement in this study is really important.**

We do not believe that it is very instructive to try and split the two (e.g. by implementing a simplified version of the RH-variability scheme). The deviations in the overall result might be large or small, but this would not lead to a lesson learned: Rather what is necessary is that parametrizations are consistent and that as few as possible empirical/tunable parameters are introduced. We add a statement on this necessity in the revised manuscript, referring to more elaborate discussions on the topic in the literature (e.g. Arakawa, 2004)).

**Comment on reply to major comment 3: I think the volume-weighted average refractive indices for internally mixed aerosols will lead to an overestimation of absorption enhancement. However, the authors describe that "the refractive indices for black carbon were updated to reduce the negative bias due to aerosol absorption enhancement" (page 3, lines 30-31). Please explain more about the consistency of these points.**

Stier et al. (2007) find that updated values of the refractive indices of BC from Bond and Bergstrom (2006) "reduce the short- wave anthropogenic aerosol top-of-atmosphere (TOA) radiative forcing clear-sky from -0.79 to -0.53 W m$-2$ (33%) and all-sky from -0.47 to -0.13 W m$^{-2}$ (72%)". Hence, the new refractive indices increase absorption enhancement. Implementing a more sophisticated mixing rule (Bruggeman, 1935) indeed decreases absorbtion enhancement. However, they find that this change is relatively small as compared to the changes that are asscociated with the updated refractive indices, namely for short- wave anthropogenic aerosol TOA radiative forcing clear-sky from -0.53 to -0.58 W m$-2$ (+9%) and for all-sky from -0.13 to -0.18 W m$-2$ (+39%).

We changed the formulation "the refractive indices for black carbon were updated to reduce the negative bias due to aerosol absorption enhancement" to "refractive indices for black carbon were updated with values from Bond and Bergstrom (2006) that led to a reduction of the negative bias due to aerosol absorption enhancement (Stier et al., 2007)".

**The description that "the absorption enhancement of BC due to mixing with hydrophilic aerosols is compensated by the lower life time and abundance of BC" (page 3, lines 31-32) should not be used in this part. This may be correct when both properties (absorption and CCN activity of BC) are changed simultaneously (e.g. aging in the real atmosphere). However, here we focus on the treatment of absorption enhancement only (BC lifetime is probably independent to the treatment of absorption enhancement in the author's model). The author's description cannot be used to support ignoring absorption enhancement in a model.**

We acknowledge the referees concerns regarding the compensating effect on RFari of a potentially strong absorbtion enhancement and remove the statement form our manuscript.

However, we can still can say that our estimates are consistent with the assumptions in the model and if better representations of absorption enhancement will be available in the future our parameterization is then able to capture the effect of subgrid-scale variability of RH on absorption enhancement. Hence, our results might be biased due to a poor representation of absorption enhancement, this is not a issue of our parameterization but should be considered in future studies. Therfore, our study underlines the importance of investagtions into a better representation of absorbtion enhancement.

**Other comments: Page 2, line 25: "They show" is better than "In this study".**

Done.

**Page 3, lines 1-2: the effect of?**

Done.

**Page 4, line 3: add "2.2 " before "Hygroscopic growth HAM2".**

Done.

**List of relevant changes**

Page and line numbers are given for the marked-up manuscript version.

**Page 1, line 4ff**

We changed the sentence

*"In this study, a stochastic parameterization of subgrid-scale variability of RHcls is applied"*

to

*"While previous studies based on limited-area models and also a global model suggest that sub-grid scale variability of RH should be taken into account for estimating RFari, here we present the first estimate of RFari using a global atmospheric model with a parameterization for sub-grid scale variability of RH that is consistent with the assumptions in the model."*

to highlight the scientific importance of our study already in the abstract.

**Page 2, line 12**

The phrase

*"Various studies show that GCMs Studies"*

was changed to

*"Studies based on limited area models suggest that GCMs may"*

to clarify that previous studies where done using limited area models.

**Page 2, line 32ff**

We added a sentence about the study by Quaas (2012) that reveals that the assumptions about the width of the PDF made in Haywood and Ramaswamy (1998) strongly differ from satellite observations.

*"This is a rather strong simplification (especially having the non-linear hygroscopic growth in mind) in comparison to findings of Quaas (2012) who suggests a change of the width of a uniform PDF from about ±20% at the surface to about ±65% in the middle troposphere."*

**Page 3, line 8ff**

We added two sentences that emphasize the importance of a consistency of parameterization with the model assumptions:

*"Hence, the parameterization complies with the necessity to be consistent and to introduce as few as possible empirical/tunable parameters. For a more elaborate discussion on the topic in the literature we refer to e.g. Arakawa (2004)."*

**Page 4, line 8f**

We changed the phrase

*"refractive indices for black carbon were updated to reduce the the negative bias due to aerosol absorption enhancement (Stier et al., 2007)"*

to

*"refractive indices for black carbon were updated with values from Bond and Bergstrom (2006) that led to a reduction of the the negative bias due to aerosol absorption enhancement (Stier et al., 2007). "*

The new wording clarifies that it was not the intention of Stier et al. (2007) to reduce absorption enhancement by updating the refractive indices for black carbon but that updated values of for the refractive indices for black carbon from Bond and Bergstrom (2006) led to a reduction of absorption enhancement.

**Page 4, line 9ff**

We removed the sentence

[revised manuscript text omitted]